# The Quality and Fertilizing Potential of Red Deer (*Cervus elaphus* L.) Epididymal Spermatozoa Stored in a Liquid State

**DOI:** 10.3390/ijms232314591

**Published:** 2022-11-23

**Authors:** Anna Dziekońska, Magdalena Koziorowska-Gilun, Władysław Kordan, Nicoletta M. Neuman, Angelika M. Kotlarczyk, Anna J. Korzekwa

**Affiliations:** 1Department of Animal Biochemistry and Biotechnology, University of Warmia and Mazury in Olsztyn, Oczapowskiego 5, 10-719 Olsztyn, Poland; 2Department of Biodiversity Protection, Institute of Animal, Reproduction and Food Research of the Polish Academy of Sciences (IAR & FR PAS), Tuwima 10 Str., 10-748 Olsztyn, Poland

**Keywords:** red deer, epididymal sperm, liquid storage, in vitro, in vivo

## Abstract

The aim of this study was to assess the quality and fertilizing potential of red deer epididymal spermatozoa stored in a liquid state for up to 11 days (D11). In Experiment 1, sperm quality was determined. In Experiment 2, the efficiency of in vitro fertilization (IVF) and artificial insemination (AI) of stored sperm were evaluated. An analysis of sperm quality on D5 of storage revealed a decrease (*p* < 0.05) in motility and morphology, and a higher proportion of apoptotic spermatozoa. On D1, D7 and D10, the total motility of sperm for IVF and AI was determined to be 82.6%, 71.0% and 64.8%, respectively. The results of IVF and AI demonstrated that the fertilizing potential of spermatozoa differs between days of storage. The percentage of blastocysts was higher when oocytes were fertilized on D1 (17.4 %) compared to D7 (8.5%) and D10 sperm (10.5%). Differences were noted in the pregnancy rates of inseminated hinds. The insemination with D1, D7 and D10 sperm led to live births (33% from D7 and D10). The results indicate that the quality of red deer epididymal spermatozoa remains satisfactory during ten days of storage in a liquid state, and that these spermatozoa maintain their fertility potential.

## 1. Introduction

Postmortem harvesting of spermatozoa and oocytes for reproductive purposes creates new opportunities for assisted reproductive technology (ART) and contributes to progress in animal breeding [1]. The preservation of epididymal spermatozoa can be the ultimate possibility for farm animals of high genetic value or for endangered species when the collection of ejaculated sperm is not possible [2,3]. 

Freshly ejaculated sperm, as well as sperm preserved in a liquid or frozen state, is most widely used for reproductive purposes in animals. However, sperm is not always easy to acquire, in particular in free-living animals and in many farmed species, including cervids. Ejaculated sperm is difficult or even impossible to collect due to the behavioral responses of these animals. For this reason, epididymal spermatozoa constitute a valuable and, in many cases, the only source of genetic material for reproductive purposes [1].

Previous studies have shown that despite certain differences in the characteristics of epididymal spermatozoa and freshly ejaculated sperm, the former can be effectively used for in vitro fertilization (IVF) and artificial insemination (AI) of farm animals, including pigs [4], bulls [5] and stallions [6]. 

The suitability of preserved sperm harvested postmortem from free-living and farmed animals, including red deer, for IVF and AI has been rarely investigated in the literature [7,8,9]. Both fresh sperm and cryopreserved sperm were analyzed in the cited studies. Cryopreservation enables long-term storage of viable sperm cells. However, this procedure is costly and time-consuming, and it decreases the fertilizing potential of the sperm [10]. This method is not always recommended because sperm collected from some individuals do not survive well after freezing [11]. These problems can be overcome in liquid storage which is a simpler and cheaper method than cryopreservation. However, the quality and fertilizing potential of the sperm have to be maintained during transport to breeding centers to guarantee successful fertilization and widespread use of this method.

Research has demonstrated that red deer epididymal spermatozoa stored in a liquid state remain functional for a relatively long time. The sperm of Iberian red deer can be stored for up to several days [12,13], and the sperm of European red deer for up to 25 days [14]. The suitability of preserved sperm for reproductive purposes has to be assessed, and such evaluations can be performed during in vitro and in vivo fertilization procedures. 

The present study was conducted on the assumption that red deer sperm can be preserved for a long time (up to 10–11 days) in a liquid state without compromising their quality and fertilizing potential. Therefore, the aim of this study was to assess the quality of red deer epididymal spermatozoa and their suitability for in vitro and in vivo fertilization. The study involved two experiments. In the first experiment, the quality of sperm stored in a liquid state was evaluated by analyzing sperm motility (CASA system), morphology, plasma membrane and acrosome integrity (fluorescent staining), mitochondrial membrane potential, DNA fragmentation and apoptotic changes. In the second experiment, sperm harvested from five stags and stored in a liquid state for one, seven and ten days were used in IVF and AI procedures.

## 2. Results

### 2.1. Experiment 1: The Effect of Storage Time (Days) on the Quality Parameters of Red Deer Epididymal Spermatozoa Stored in a Liquid State

The motility analysis demonstrated that more than 60% of red deer epididymal spermatozoa remained motile up to the tenth day of storage in a liquid state at a temperature of 5 °C (Figure 1A). A significant decrease (*p* < 0.05) in TMOT, PMOT, VAP and VSL values was observed on D5, and a subsequent decrease was noted on D11 (Figure 1A–D, respectively). A significant decrease in VCL, STR and LIN values was observed only on D11 (Figure 1E,H,I, respectively). No significant changes (*p* > 0.05) in ALH and BCF were noted throughout the entire storage period (Figure 1F,G, respectively). 

The results of the microscopic analysis of various cellular structures are presented in Table 1. The evaluation of acrosome integrity by Giemsa staining and fluorescent staining with FITC/PNA labeling and the viability analysis revealed a significant decrease (*p* < 0.05) in the studied parameters on D10. Significant changes in the percentage of sperm cells with normal morphology and an increase (*p* < 0.05) in the percentage of sperm cells with abnormal tails were noted on D5. The percentage of sperm with morphological defects of the head and the midpiece increased significantly on D11. The percentage of sperm with active mitochondria decreased significantly (*p* < 0.05) on D5. DNA integrity did not differ significantly on different days of storage. A significant increase (*p* < 0.05) in the percentage of apoptotic sperm was noted on D5. The percentage of viable sperm without apoptotic changes decreased significantly (*p* < 0.05) on D5, and on D10, it was 20% lower than on D1.

### 2.2. Experiment 2: Fertilizing Potential of Red Deer Epididymal Spermatozoa Stored in a Liquid State

Fertilizing potential of red deer epididymal spermatozoa stored in a liquid state is presented in Table 2. We collected 115, 83 and 44 oocytes predicted for IVF with sperm stored from D1, D7 and D10, respectively. The maturation process was successful for all oocytes and it ranged from 90 to 94% between the groups (*p* > 0.05). The oocyte fertilization rates varied between the groups, but no significant differences were found. Cleavage patterns (development from the zygote to the morula) were observed in 56.5% of the oocytes fertilized with D1 sperm, 27.1% with D7 and 13.2% with D10. Significant differences were noted between oocytes fertilized with D1 sperm and D7 and D10 sperm (*p* < 0.05). Blastocysts rate ranged from 8.5 to 17.4% between groups and it did not differ significantly (*p* > 0.05).

In all hinds, sperm was successfully deposited in the cervix during the insemination and repeat insemination. Pregnancy rates in artificially inseminated hinds varied considerably at 50% with D1 sperm, 33% with D7 sperm and 100% in hinds with D10 sperm. Intracervical insemination of six females with sperm stored for one day (group 1) resulted in three pregnancies that were confirmed by progesterone levels, and two pregnancies by the PAGs test. In turn, the insemination of six females with D7 sperm (group 2) and D10 sperm (group 3) resulted in two and six pregnancies, respectively, that were confirmed by both tests. However, the birth rates for D1 and D10 were lower than the pregnancy rates. 

## 3. Discussion

The use of red deer epididymal spermatozoa for IVF and AI remains insufficiently investigated in the literature [8,9,15], and most studies examined freshly ejaculated and cryopreserved sperm. The suitability of long liquid-stored epididymal spermatozoa for both procedures has never been studied. Therefore, this is the first study to evaluate the applicability of liquid-stored sperm for reproductive purposes. This is an important consideration since in farm-raised animals, ejaculated sperm is difficult to obtain and cryopreserve, which is why the liquid storage of epididymal spermatozoa can resolve many technical and financial problems. 

The results of the analyses assessing the quality of red deer epididymal spermatozoa (including motility and plasma membrane integrity) stored in Bovidyl extender at a temperature of 5 °C for 11 days confirmed our previous observations of red deer sperm stored in Salomon’s extender [14]. The above extenders have a similar composition and offer similar conditions for sperm storage. Before Bovidyl was used in this study, red deer sperm stored in Bovidyl and Salomon’s extenders had been checked for significant differences in motility and viability, and no such differences had been found. 

Sperm motility decreased gradually on successive days of storage, and the earliest significant changes were observed in motility parameters (TMOT, PMOT, VAP, VSL). Sperm kinetic parameters such as VAP, VCL and VSL are particularly important in assessing the fertility of red deer stags [16,17]. In the current study, VCL values remained high until D10, whereas VAP values decreased gradually, which could suggest that the fertilizing potential of spermatozoa can be fully preserved during storage. Ramon et al. [18] demonstrated that VCL and VAP values of red deer sperm with high fertilizing potential remained high throughout storage. In this study, VAP and VSL values decreased, while insignificant changes in VCL values up to D10 of storage indicated sperm circulation, which may signal sperm dysfunction and suitability for fertilization.

In this study, the percentage of sperm with normal apical ridge acrosomes (NAR) and normal plasma membranes decreased very slowly, which could indicate that these structures are quite resilient to various storage conditions. Similar findings were reported by other authors [19,20]. However, differences in acrosome integrity were observed between the applied analytical methods, where fluorescent staining was more sensitive than Giemsa staining. 

The morphological analysis revealed a significant decrease in the percentage of sperm with normal morphology on the fifth day of storage, which resulted mainly from considerable tail damage. The morphological analysis did not account for cytoplastic droplets which are commonly observed in epididymal spermatozoa [21]. The changes in sperm morphology could indicate that the tails of epididymal spermatozoa are more sensitive to cooling than the heads. Similar observations were made by Ros-Santaella et al. [16] in red deer sperm subjected to freezing and thawing.

The energy for sperm motility is generated by glycolysis and oxidative phosphorylation, and motility characteristics can be affected by disruptions in these processes [17,22]. Oxidative phosphorylation generates the highest amount of energy for sperm motility, and this process can be disrupted by a decrease in MMP. Therefore, a significant decrease in MMP on the fifth day of storage considerably decreased sperm motility. Despite the above, more than 70% of sperm had active mitochondria throughout the entire storage period (up to 11 days), which enabled effective energy production and motility. Similar results were noted in a previous study [14]. A decrease in MMP is generally accompanied by an increase in the percentage of sperm with apoptotic changes [23]. Apoptotic changes are intensified during storage, which was also noted in the present study. On D5 of storage, the percentage of apoptotic sperm increased relative to D1, whereas significant differences were not observed in the viability analysis. These results indicate that apoptosis occurs much earlier than changes in sperm plasma membrane, which corroborates the findings of Martínez-Pastor et al. [23]. 

The integrity of nuclear DNA, which is largely influenced by chromatin integrity, is a key determinant of the fertilizing potential of sperm. DNA damage is regarded as one of the factors that decrease sperm’s ability to bind to ovarian surface epithelium [24], compromise embryonic development [25] and contribute to embryonic and fetal mortality [26]. Therefore, DNA integrity plays an important role in assessments of sperm’s fertilizing potential. In the current study, a significant decrease in DNA integrity was not observed during the entire period of sperm storage in a liquid state, which suggests that DNA remained relatively stable. Similar observations were made in ejaculated boar sperm stored in a liquid state [27] and in the epididymal spermatozoa of mice [28] and Iberian red deer [13].

In this study, a comprehensive analysis of epididymal spermatozoa revealed that sperm viability and functionality are preserved for a long time during liquid storage. Nonetheless, the sperm’s ability to fertilize an egg cell in vitro and, particularly, in vivo is the only reliable measure of their biological value.

In the second experiment, the fertilizing potential of liquid-stored epididymal spermatozoa for IVF and AI treatments was assessed in sperm samples collected from five of the eleven analyzed stags. These spermatozoa were characterized by the highest initial motility (approx. 80%) and satisfactory motility on D7, D10 and D11 (over 60%). An analysis of the quality of stored sperm that were used for IVF and AI revealed a similar relationship to the above analysis (see Appendix A). Sperm samples for IVF and AI were free of microbiological contamination that could compromise the fertilizing potential of stored sperm and embryo development. 

The results of the IVF procedure demonstrated that both sperm stored for one day and sperm stored for seven and ten days were able to fertilize mature oocytes in vitro, and that their fertilizing potential was maintained. However, differences between experimental groups of hinds were noted in the number of harvested and cultured mature oocytes that were suitable for IVF, which may be due to individual differences among female deer. Nevertheless, maturation of the oocytes was very effective regardless of the experimental group (90–94%). In cows, about 92% [29] of oocytes matured after 24 h, whereas in deer this was 69.4 % [30]. Our result proves that the conditions of culture were optimally adjusted. However, a decrease was observed in cleaved embryos and blastocyst rates compared to the number of fertilized oocytes in our study. Our earlier study showed blastocyst rates ranging between 12.3% and 21.8% dependent on the reproductive stage of the hind which was the oocytes’ donor [31]. There is no doubt that the pharmacological method of the estrus cycle synchronization affects both the number of oocytes retrieved and the embryo development. These differences could be also attributed to a decrease in the fertilizing potential of sperm caused by storage and aging, as mentioned above. Previous studies demonstrated that apoptotic changes can decrease the fertilizing potential of bull sperm [32]. Apoptotic sperm can be also responsible for the loss of human embryos in the early stages of development [33]. Therefore, the highest number of cleaved embryos and blastocysts we measured was in the group fertilized with sperm stored for one day, and the results were lower than those reported by other authors who conducted IVF with cryopreserved sperm [34,35]. Thus, storage-induced changes in sperm, embryonic development affected by oocyte quality, in vitro culture conditions such as temperature, and the applied culture media [36,37] are all aspects demanding further study.

In this study, considerable differences were found in the success rates of AI involving the same sperm samples. Interestingly, AI was more effective in group 3 (all hinds were pregnant) than in groups 1 and 2 (3 and 2 out of 6 hinds were pregnant, respectively). Similar to IVF, these differences could be attributed to variations in the analyzed hinds’ responses to estrus synchronization and irregular ovulation in groups 1 and 2. The time between eCG injection, estrus and insemination must also be accurately defined for successful estrus synchronization [11,31]. Artificial insemination success rates, including pregnancy rates, were satisfying in this study, which indicates that the fertilizing potential of epididymal spermatozoa is fully preserved during ten–eleven days of storage. This was evidenced by the fact that hinds inseminated with stored sperm gave birth to live calves. However, the number of live born calves was far lower than might be expected based on the noted pregnancy rates. The causes of fetal mortality in this study are not clear, but one reason could be a gradual deterioration in the quality of epididymal sperm during storage. Besides sperm and oocyte quality, the effectiveness of fertilization is influenced by factors such as the performance of the AI procedure, including the maintenance of appropriate hygienic conditions and the stress associated with manipulation. Pathogen infections may have influenced embryo development and fetal mortality [38,39,40]. In hinds, especially in group 3, the presence of *Trueperella pyogenes* bacteria was detected in vaginal discharge, which could have contributed to the significant fetal mortality [41]. Despite the above, birth rates were similar to hinds inseminated with stored epididymal sperm (33%) and to hinds inseminated with fresh and cryopreserved sperm (˂40%) presented in other reports [2,11]. However, the obtained results relate to small-sized experimental groups and therefore this study can only be considered as preliminary and will be continued.

## 4. Materials and Methods

The diagram of the experiments carried out in this study is presented in Figure 2.

### 4.1. Animals

The present study was conducted on red deer (hybrids of various subspecies with a predominance of European deer), and it involved mature males (*n* = 11; 4–5 years; weighing about 145 kg) and mature females (*n* = 36; 3–4 years; weighing about 95 kg) kept in a deer farm in Rudzie near Gołdap (north-eastern Poland). In spring and fall, the animals were fed only a pasture-based diet, and in winter, their diet was supplemented with silage and cereal grain (10 kg of silage/day/animal + 1 kg of cereal grain). 

### 4.2. Collection and Initial Evaluation of Epididymal Spermatozoa 

Spermatozoa were obtained from the epididymides of 11 stags that were legally culled during the rutting season (September). Sperm samples were collected directly (up to 0.5–1 h postmortem) from the cauda epididymis into Eppendorf tubes, according to the method described earlier by Dziekońska et al. [14]. The samples were assessed for motility and concentration. Sperm motility was determined subjectively. Samples with sperm motility higher than 70% were used in further analyses. The concentration of spermatozoa was determined in a Bürker counting chamber (Equimed-Medical Instruments, Kraków, Poland). 

### 4.3. Liquid Storage of Sperm 

Sperm samples were diluted with Bovidyl extender (Bovidyl^®^, Minitub GmbH, Tiefenbach, Germany) containing TRIS, BSA, citric acid, sugars, buffers, proprietary components, antibiotics, double distilled water and 10% clarified egg yolk at room temperature. The samples (100 × 10^6^ spermatozoa/mL) were placed in 50 mL plastic tubes, immediately transferred to a portable refrigerator (5 °C) and transported to the laboratory of the Department of Animal Biochemistry and Biotechnology of the University of Warmia and Mazury in Olsztyn. Sperm samples were stored at a temperature of 5 °C throughout the experiment (11 days). 

### 4.4. Experiment 1: Sperm Quality Analysis 

Sperm quality was analyzed on the day the samples were transported to the laboratory (D1) and then on days 5 (D5), 7 (D7), 10 (D10) and 11 (D11). 

#### 4.4.1. Analysis of Sperm Motility Parameters 

Sperm motility parameters were assessed using the computer-assisted sperm analysis (CASA) system (Hamilton-Thorne Research, HTR, IVOS version 12.3; Beverley, MA, USA). The IVOS settings were applied according to the Hamilton Thorne technical guide v. 12.3 for gazelle/deer [14]. Sperm samples (100 × 10^6^ spermatozoa/mL) were diluted 1:4 with phosphate buffered saline (PBS) and incubated at 37 °C for 20 min. Then, the samples (5 μL) were placed in a pre-warmed Makler counting chamber (Sefi-Medical Instruments Ltd., Haifa, Israel) and examined at 37 °C. A minimum of five fields per sample were assessed, with approximately 200 spermatozoa per field. The CASA-determined sperm motility parameters included total motility (TMOT, %), progressive motility (PMOT, %), curvilinear line velocity (VCL, μm/s), straight line velocity (VSL, μm/s), average velocity (VAP, μm/s), amplitude of lateral head displacement (ALH, μm), beat cross frequency (BCF, Hz), linearity coefficient (LIN, %) and straightness (STR, %). The spermatozoa were considered motile with VAP > 21.9 and VSL > 6.0 μm/s, and progressive with VAP > 75.0 μm/s and STR > 80%.

#### 4.4.2. Sperm Morphology Assay

Smears were prepared from sperm samples diluted with PBS (1:4). The smears were air-dried and stained by the Giemsa staining method according to Watson [42]. The morphological features of 200 sperm cells from each sample were evaluated under a phase-contrast microscope (1000× magnification). The percentage of sperm with normal morphology (%, MOR), the percentage of sperm with normal apical ridge acrosomes (%, NAR) and the percentage of sperm with morphological defects of the head, midpiece and tail (%) were determined. 

#### 4.4.3. Fluorescence Assays

The acrosomal status of spermatozoa was assessed by the fluorescence method using FITC-labeled peanut agglutinin (FITC-PNA; Sigma, Saint Luis, MO, USA), as described in a previous study [14] with some modifications. Sperm samples (20 μL, 100 × 10^6^ spermatozoa/mL) were extended in 180 μL of PBS solution and incubated with 2 μL of FITC-PNA solution (2 mg FITC-PNA in 1 mL PBS) at 37 °C for 5 min. Then, 2 μL of propidium iodide solution (PI, 2.4 μM in Tyrode’s salt solution) was added to the samples and incubated at 37 °C for 5 min. Finally, 10 μL of 10% formalin solution was added to the stained cells. The samples were evaluated under a fluorescence microscope (Olympus BX 41, Tokyo, Japan) at 600× magnification. A minimum of 200 cells per slide were examined in each aliquot. The results were expressed as the percentage of viable sperm cells with intact acrosomes (non-stained spermatozoa in the head region; FITC-PNA^−^/PI^−^). 

The plasma membrane integrity of sperm was assessed using dual fluorescent probes, SYBR-14 and propidium iodide, PI (Live/Dead Sperm Viability Kit; Molecular Probes, Eugene, OR, USA), according to the method described previously by Garner and Johnson [43] with some modifications. Sperm samples (20 μL, 100 × 10^6^ spermatozoa/mL) were extended in 180 μL of PBS solution. Subsequently, the samples were incubated with 2 μL 1 mM SYBR-14 solution in HEPES-BSA solution (pH 7.4) and 2 μL PI (2.4 μM in Tyrode’s salt solution) at 37 °C for 10 min. Aliquots of the stained sperm cells were examined using a fluorescence microscope (Olympus BX 41, Tokyo, Japan) at 600× magnification. A minimum of 200 spermatozoa were counted per slide. The results were expressed as the percentage of viable sperm cells (sperm with bright green fluorescence; SYBR^+^/PI^−^).

The mitochondrial membrane potential of sperm was assessed using JC-1 (Molecular Probes, Eugene, OR, USA)/PI dual fluorescent probes, according to a previously described method [14] with some modifications. Briefly, sperm samples (20 μL, 100 × 10^6^ spermatozoa/mL) were extended in 180 μL of PBS solution and incubated with 1 μL of JC-1 solution (1 mg JC-1/mL anhydrous dimethyl sulfoxide, DMSO) for 15 min at 37 °C. Then, sperm samples were stained with 2 μL PI for 5 min at 37 °C. Sperm cells displaying orange–red fluorescence in the mid-piece region were considered as viable spermatozoa with high MMP, whereas sperm cells displaying red fluorescence in the head and green fluorescence in the mid-piece region were classified non-viable spermatozoa with low MMP. A minimum of 200 cells per slide were evaluated using a fluorescence microscope (Olympus BX 41, Tokyo, Japan) at 600× magnification. The results were expressed as the percentage of viable sperm cells with high MMP.

Apoptotic changes in sperm were assessed with the Vybrant Apoptosis Assay Kit #4 (Molecular Probes Inc., Eugene, OR, USA), according to the method described by Trzcińska and Bryła [44] with some modifications. A 200 μL sperm sample (10 × 10^6^ sperm/mL) was first combined with 2 μL JC-1 (100 μM) to facilitate the identification of sperm not stained with YO-PRO-1 or PI, and incubated for 10 min at 37 °C. Then, 2 µL of YO-PRO-1 (100 µM) and 2 µL of PI (2 µM) were added and incubated again for 10 min at 37 °C. Aliquots of stained sperm cells were examined under a fluorescence microscope (Olympus BX 41, Tokyo, Japan) at 600× magnification. A minimum of 200 cells per slide were examined in each aliquot, and four subpopulations were identified in the assay (Figure 3A,B): viable sperm categorized as negative for both YO-PRO-1 and PI and positive for JC-1 (YOPRO^−^/PI^−^); sperm with apoptotic-like changes in the plasma membrane were categorized as positive for YO-PRO-1 and JC-1 but negative for PI (YOPRO^+^/PI^−^); moribund/dead sperm were positive for YO-PRO-1, PI and JC-1; and dead sperm were positive for PI.

DNA integrity was assessed with acridine orange (AO; Life Technologies Ltd., Grand Island, NY, USA), according to the method described by Partyka et al. [45] with some modifications. A suspension of sperm samples (100 μL) was subjected to brief acid denaturation with 200 μL of a lysis solution (Triton X-100 0.1% (*v*/*v*), NaCl 0.15 M, HCl 0.08 M, pH 1.4). The samples were left for 30 s in the dark, and 600 µL of AO solution (6 µg AO/mL in a buffer: citric acid 0.1 M, Na2HPO4 0.2 M, EDTA 1 mM, NaCl 0.15 M, pH 6) was added. The samples were left in the dark for 3 min, after which a minimum of 200 sperm cells were randomly selected under a fluorescence microscope (Olympus BX 41) at 600× magnification. In the analysis, the double-stranded sperm subpopulation (%, DNA integrity) with green fluorescence was distinguished from the sperm population with damaged DNA and yellow to red fluorescence (moderately and completely denatured DNA, respectively). 

### 4.5. Experiment 2: Assessment of the Fertilizing Potential of Epididymal Spermatozoa 

The fertilizing potential of epididymal spermatozoa stored for one, seven and ten days in Bovidyl extender was determined by IVF of the oocytes collected from hinds and by AI of hinds. Sperm samples collected from five males were used for IVF. The samples for AI were pooled from the same males. The samples collected from each male were assessed for quality according to a previously described procedure. The motility of sperm was assessed subjectively before IVF and AI procedures.

The hinds used in both procedures were pharmacologically treated for induction of estrus and ovulation according to the method described by Korzekwa et al. [31].

#### 4.5.1. Induction of Estrus and Ovulation and Oocyte Collection

Ovaries were collected postmortem from 18 red deer hinds (from three groups of six hinds each) directly after slaughter at a deer farm in Rudzie near Gołdap (north-eastern Poland) on the 4th day of the estrous cycle after pharmacological synchronization of the animals. Estrus and ovulation were induced during the estrous cycle (September) by applying a single controlled-release vaginal sponge for sheep (Chronogest, MSD Animal Health, Milton Keynes, UK, 20 mg flugestone acetate). For better synchronization, the sponge was removed after 14 days, and equine serum gonadotropin (eCG, Syncrostim, 250 I, Ceva Animal Health, Warsaw, Poland; 250 IU) was injected intramuscularly [31]. Estrus was observed 54–56 h after the injection. The day of the estrous cycle was determined based on macroscopic observations of the ovaries and the uterus, and it was confirmed by measuring plasma 17-beta estradiol (E2) and progesterone (P4) levels by a radioimmunoassay. Blood was sampled from the heart. The farmed animals were culled for financial reasons and to renew the herd. The ovaries were transported to the laboratory in sterile phosphate-buffered saline. Cumulus–oocyte complexes (COCs) were obtained by aspiration from subordinate ovarian follicles measuring less than 5 mm in diameter, and by maceration of ovarian tissue from the same ovary after aspiration. A stereo microscope (Discovery V20, Zeiss, Poznan, Poland) was used to identify COCs consisting of oocytes with homogeneous ooplasm without dark spots, surrounded by at least three layers of compact cumulus cells. The COCs were washed twice in a wash medium (61008; IVF Bioscience, Falmouth, UK) and subsequently in a maturation medium (61002; IVF Bioscience, Falmouth, UK). 

#### 4.5.2. In Vitro Oocyte Maturation and In Vitro Fertilization 

In vitro oocyte maturation (IVM) and in vitro fertilization were conducted according to method described by Korzekwa et al. [31]. Groups of 20 immature COCs were placed in Petri dishes containing 100 µL of the maturation medium covered with mineral oil (M8410, Sigma, Poznań, Poland) and were incubated at 38.5 °C in a 5% CO_2_ humidified air atmosphere for 23 h. The COCs were washed in a fertilization medium (61003; IVF Bioscience, Falmouth, UK).

The COCs were divided into three experimental groups: group 1 was fertilized with the sperm stored in a liquid state at 5 °C for one day (*n* = 6), and group 2 was fertilized with the sperm stored in a liquid state at 5 °C for seven days (*n* = 6) and group 3 was fertilized with the sperm stored in a liquid state at 5 °C for ten days (*n* = 6). Before IVF, sperm was incubated in a capacitation medium (61004; IVF Bioscience, Falmouth, UK) for 24 h, and motile spermatozoa were recovered by the swim-up procedure. After incubation, sperm was double centrifuged at 200× *g* for 5 min, the supernatant was removed, and the sperm pellet was diluted in an appropriate volume of the fertilization medium to a final concentration of 2 × 10^6^ motile sperm/mL. Groups of 20 COCs were co-incubated with spermatozoa in Petri dishes containing 100 µL of the fertilization medium, covered with mineral oil for 15 h at 38.5 °C in a 5% CO_2_ humidified air atmosphere. Embryos (blastocysts) were collected between day 6 and day 9. The developmental stage and quality of the embryos were determined under a microscope (Zeiss Axio Observer A1; Carl Zeiss Microscopy GmbH, Jena, Germany; magnification 200×) based on the International Embryo Transfer Society manual. The quality of blastocysts was scored as follows: grade A—excellent; grade B—good; grade C—fair and moderate; grade D—poor; and grade E—dead or degenerating (Figure 4). Embryos classified into quality grades A–C were used to determine the rate of embryo development to the blastocyst stage (IVF success rate) and were assessed for morphological quality.

#### 4.5.3. Artificial Insemination (AI) Procedure

The AI procedure involved 18 hinds (three groups of six hinds each) whose estrous cycle had been previously synchronized (as described above). The first group was inseminated with sperm stored for one day, the second group with sperm stored for seven days and the third group with sperm stored for ten days. The hinds were inseminated twice: 56 h (first insemination) and 68 h after the administration of eCG (using to repeat insemination sperm stored for 2, 8 and 11 days, respectively). During the AI procedure, hinds were physically immobilized in a squeeze chute without anesthesia. The vulvar region was aseptically prepared, and a catheter for intracervical insemination (IMV Technologies, L’Aigle, France) was introduced into the cervix. A digital rectal examination was performed to facilitate catheter placement. Around 250 µL of the sperm suspension (containing around 25–30 × 10^6^ of spermatozoa) heated to 35–37 °C was placed in a syringe, injected into the catheter and deposited in the cervix. The success rate of fertilization with epididymal sperm was determined by dividing the number of pregnant hinds by the number of inseminated hinds, and it was expressed as percentage. Additionally, the birth rate for hinds was calculated by dividing the number of calves by the number of inseminated hinds. The results were expressed as a percentage.

#### 4.5.4. Diagnosis of Pregnancy

Thirty days after insemination, peripheral blood was sampled from the jugular vein to determine the levels of pregnancy-associated glycoproteins (PAGs) and progesterone (P4). PAGs and P4 concentrations were determined according to the method described by Korzekwa et al. [31]. 

Serum levels of PAGs were determined with the EIA Pregnancy Test (99-41169, IDEXX, Westbrook, ME, USA) after validation. The average intra- and inter-assay coefficients of variation (CVs) were determined at 8.0% and 7.9%, respectively. The presence or absence of PAGs in each sample was validated with the use of the provided formula. The circulating concentrations of PAGs were calculated based on absorbance, where absorbency below 0.3 was indicative of non-pregnancy and absorbance above 0.3 was indicative of pregnancy. 

Plasma P4 levels were determined by a radioimmunoassay (RIA) (KIP1458, DIAsource ImmunoAssays, Louvain-la-Neuve, Belgium). The standard curve range was 0.12 to 36 ng/mL, and the ED50 of the assay was 0.06 ng/mL. The intra- and inter-assay CVs were 6.5% and 8.6%, respectively. Hinds with progesterone concentration of 6 ng/mL and higher were considered pregnant [46].

### 4.6. Statistical Analysis

Data were examined by repeated measure ANOVA, using the general linear model (GLM) procedure from Statistica software package, version 12.5 (StatSoft Incorporation, Tulsa, OK, USA). Shapiro–Wilk and Levene’s tests were applied to check the normality of data distribution and the homogeneity of variance, respectively. Data were not normally distributed, and they were transformed accordingly. Percentage data were subjected to arcsine transformation, and motility parameters (VCL, VSL, VAP, ALH and BCF) were log-transformed to obtain normally distributed data before performing one-way analysis of variance (ANOVA). The results were expressed as the mean ± SEM. Significant differences between means were determined by the Student–Newman–Keuls post hoc test at a significance level of *p* < 0.05. 

Fertilized oocytes, cleavage and blastocyst rates (%) were analyzed by the Fisher’s exact test. Differences were considered statistically significant at *p* < 0.05.

## 5. Conclusions

The results of this study indicate that the quality and fertilizing capacity of red deer epididymal spermatozoa are effectively preserved during liquid storage (for up to 10–11 days) and could be used in ART. However, further research, including in more animals, is needed to confirm these findings. The results of this study may indicate possible practical applications of liquid-stored epididymal sperm, which could contribute to breeding progress in cervid farms. Liquid storage is a less demanding method than cryopreservation, therefore it can be easily applied in cervid farms and can contribute to the exchange of genetic material.

## Figures and Tables

**Figure 1 ijms-23-14591-f001:**
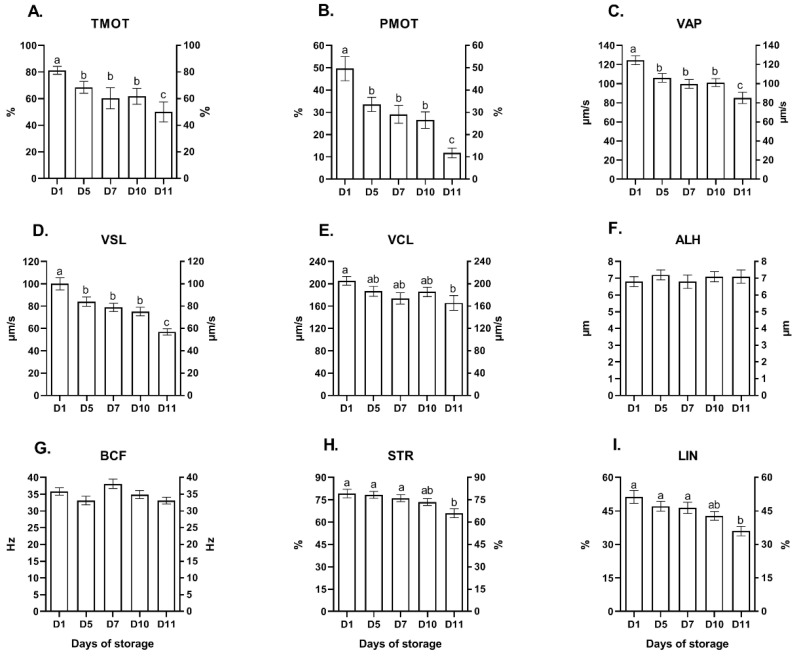
The effect of storage time (days) on motility variables of red deer epididymal sperm (mean ± SEM) stored in a liquid state at 5 °C. (**A**) TMOT, total motility. (**B**) PMOT, progressive motility. (**C**) VAP, velocity average path. (**D**) VSL, velocity straight line. (**E**) VCL, curvilinear velocity. (**F**) ALH, mean amplitude of lateral head displacement. (**G**) BCF, beat cross frequency. (**H**) STR, straightness. (**I**) LIN, linearity. Bars with different letters denote differences (*p* < 0.05) between days of storage.

**Figure 2 ijms-23-14591-f002:**
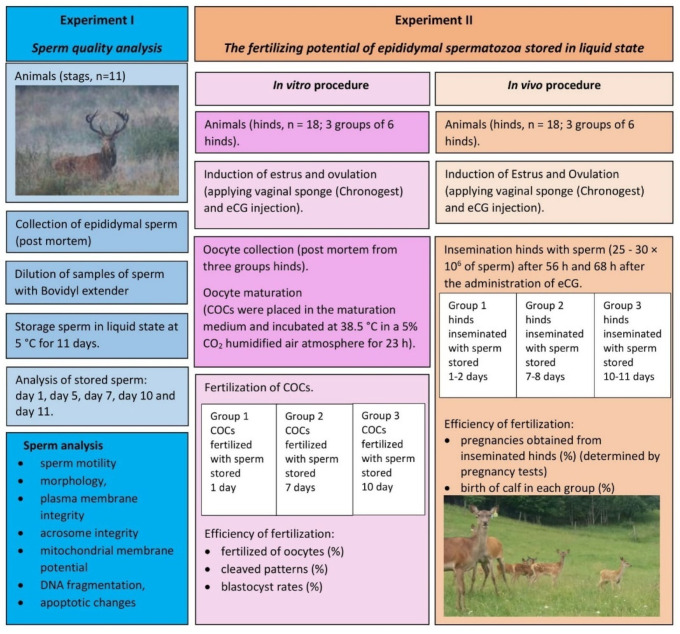
The diagram of the experiments.

**Figure 3 ijms-23-14591-f003:**
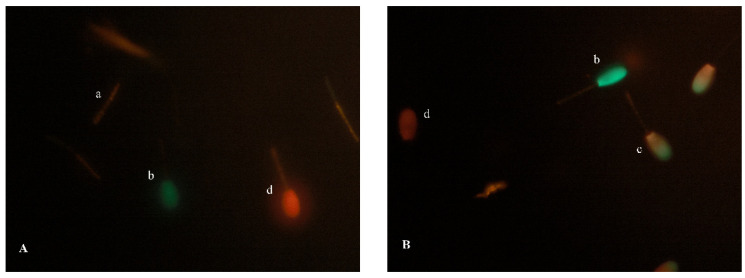
(**A**,**B**) Red deer sperm stained with the fluorescent combination of JC-1 and YOPRO/PI fluorochromes under a fluorescence microscope: (**a**) Non-apoptotic cells, yellow/orange fluorescence in midpiece (YOPRO^−^/PI^−^). (**b**) Apoptotic cell, green fluorescence in head region (YOPRO^+^/PI^−^). (**c**) Moribund/dying sperm, cell with dual fluorescence green and red in head region (YO-PRO^+^/PI^+^). (**d**) Dead sperm, red fluorescence in head region (PI^+^). Magnification 600×.

**Figure 4 ijms-23-14591-f004:**
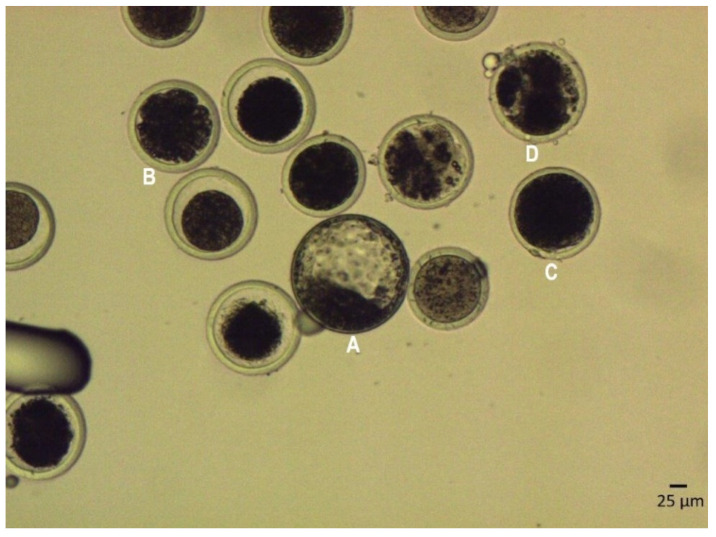
The developmental stage and quality of red deer embryos. (**A**) Blastocyst; complete blastocyst; the cavity fills almost the entire embryo; the trophoblast composed of numerous densely arranged cells that form a coherent layer. (**B**) Morula, embryo with equal blastomeres and several unequal blastomeres, incomplete compaction. (**C**) Unfertilized oocyte. (**D**) Degenerated embryos. The image was taken under phase contrast microscopy at 200× magnification. Scale bar, 25 µm.

**Table 1 ijms-23-14591-t001:** The effect of storage time (days) on the quality parameters of red deer epididymal spermatozoa (*n* = 11) stored in a liquid state at 5 °C.

Sperm Quality Parameters	Storage Time (Days)				
	Day 1	Day 5	Day 7	Day 10	Day 11
MOR (%)	75.6 ± 2.6 ^a^	67.6 ± 2.8 ^b^	66.8 ± 2.6 ^b^	63.5 ± 3.1 ^bc^	56.3 ± 3.8 ^c^
NAR (%)	89.0 ± 1.4 ^a^	85.8 ± 1.4 ^ab^	84.8 ± 1.4 ^ab^	81.2 ± 1.6 ^b^	79.8 ± 2.3 ^b^
HD (%)	5.5 ± 1.6 ^a^	6.2 ± 1.7 ^a^	7.5 ± 0.9 ^a^	7.1 ± 2.1 ^a^	20.9 ± 2.8 ^b^
MD (%)	1.7 ± 0.8 ^a^	1.3 ± 0.4 ^a^	1.2 ± 2.9 ^a^	1.1 ± 0.3 ^a^	9.2 ± 2.7 ^b^
TD (%)	7.0 ± 1.0 ^a^	14.8 ± 2.9 ^b^	17.8 ± 0.1 ^b^	17.1 ± 2.1 ^b^	21.2 ± 7.2 ^b^
PNA^−^/PI^−^ (%)	79.8 ± 3.8 ^a^	79.1 ± 1.8 ^a^	77.0 ± 4.1 ^a^	70.9 ± 3.4 ^b^	60.7 ± 5.4 ^b^
SYBR^+^/PI^−^ (%)	87.9 ± 1.8 ^a^	85.8 ± 1.4 ^a^	86.9 ± 2.2 ^a^	81.2 ± 1.6 ^b^	79.8 ± 2.3 ^b^
MMP (%)	88.6 ± 1.7 ^a^	83.1 ± 1.4 ^b^	86.0 ± 2.6 ^ab^	78.4 ± 2.3 ^b^	77.9 ± 5.5 ^b^
AO (%)	95.3 ± 0.9	91.8 ± 2.4	93.8 ± 1.3	92.0 ± 1.8	90.2 ± 1.3
YOPRO^−^/PI^−^ (%)	86.6 ± 2.1 ^a^	70.5 ± 3.7 ^b^	71.4 ± 4.3 ^b^	66.6 ± 3.5 ^bc^	58.6 ± 4.0 ^c^
YOPRO^+^/PI^−^ (%)	3.2 ± 1.0 ^a^	12.4 ± 5.4 ^b^	12.1 ± 5.6 ^b^	10.5 ± 5.1 ^b^	16.3 ± 5.4 ^b^

MOR, normal sperm. NAR, normal apical ridge acrosomes. HD, head defects. MD, midpiece defects. TD, tail defects. PNA^−^/PI^−^, acrosomal status. SYBR^+^/PI^−^, viability. MMP, mitochondrial membrane potential. AO, DNA integrity. YOPRO^−^/PI^−^, viable, non-apoptotic sperm. YOPRO^+^/PI^−^, apoptotic sperm. Values are expressed as means ± SEM. Differences between means were determined by Student–Newman–Keuls post hoc test. ^a–c^ Values within a row with different superscripts differ significantly at *p* < 0.05.

**Table 2 ijms-23-14591-t002:** Percentage of motile sperm, in vitro success rates after fertilization oocytes, and artificial insemination rates in hinds inseminated (*n* = 18) with liquid-stored epididymal sperm of red deer.

	Fertilization Potential		
	Sperm Stored One Day (Group 1)	Sperm Stored Seven Days (Group 2)	Sperm Stored Ten Days (Group 3)
TMOT (%)	82.6 ± 1.7 ^a^	71.0 ± 4.1 ^b^	64.8 ± 3.7 ^b^
PMOT (%)	42.8 ± 3.3 ^a^	34.8 ± 4.2 ^b^	21.6 ± 4.5 ^b^
Total oocytes	115	83	44
Immature oocytes	7	8	4
Mature oocytes	108/115 (93.9%)	75/83 (90.4%)	40/44 (90.9%)
Fertilized oocytes	69/108 (63.9%)	59/83 (71.1%)	38/44 (86.4%)
Cleaved embryos	39/69 (56.5%) ^a^	16/59 (27.1%) ^b^	5/38 (13.2%) ^b^
Blastocysts	12/39 (17.4%)	5/16 (8.5%)	4/38 (10.5%)
Pregnancy rate	3/6 (50%)	2/6 (33%)	6/6 (100%)
Birth rate	1/6 (17%)	2/6 (33%)	2/6 (33%)

TMOT, total motility. PMOT, progressive motility. Mature oocytes, number of matured oocytes from total oocytes (%). Fertilized oocytes, number of fertilized of oocytes from the total number of matured oocytes (%). Cleaved patterns, number of morula stage embryos obtained from fertilized oocytes (%). Blastocyst stage, number of blastocyst obtained from embryos that had reached the morula stage (%). Pregnancy rate, number of pregnancies obtained from inseminated hinds (%). Birth rate, number of calves born alive obtained from inseminated hinds (%). Values for TMOT and PMOT are expressed as means ± SEM (*n* = 5). Differences between means were determined by Student–Newman–Keuls post hoc test. Differences between groups for mature oocytes, fertilized oocytes, cleaved embryos and blastocysts were determined by Fisher’s exact test. ^a,b^ Values within a row with different superscripts differ significantly at *p* < 0.05.

## Data Availability

This report is available as a preprint on Research Square Platform, DOI: 10.21203/rs.3.rs-1555742/v1.

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
