# Peer review of "The Quality and Fertilizing Potential of Red Deer (Cervus elaphus L.) Epididymal Spermatozoa Stored in a Liquid State"

_ijms, 2022, doi:10.3390/ijms232314591_

Round 1
Reviewer 1 Report
Dear authors,
Congratulations about your experiments and manuscript. Our observations are only to contribute with your paper.
Lines 64-67: Spermatozoa analysis, but without mitochondrial test. If the authors putting all evaluations in this section, can be include this test.
Lines 73-79: The results are not significant? There aren't "p" description. If no significant, put, please, p >0.05.
Line 140: I didn't understand this phrase.
Line 216: The authors used "fresh sperm". Fresh sperm, in our knowledge, it's a recent semen collected. After collection, dilution, and temperature reduction, this sample is cryopreserve, no?
Line 220: repetitive expression. Our sugesttion: "This may be due to individual differences among females deer."
About this point, my sight is the fertilization is a multifactorial condition that depend too oocyte quality, spermatozoa quality and, a third factor - quality of AI service (since hygiene procedure until manipulation stress).
Line 395: PMSG is a older term. Our suggestion is remove this information.
Line 425: The authors related: "Embryos (blastocysts) were collected between day 6 and day 9." This phrase connote that embryos were collected, probably, in the in vivo process. However, the embryos were cultivated in vitro and selected between daty 6 and 9 for transfer to deer females.
Line 447: one day or two day?
And, in addition, in my perception about the kinetics parameters, the reduction of VSL and maintenance of VCL indicate probably spermatozoa circular movement, which may be a signal to reduction of sperm function, i.e., there are fertilization but decrease on embryonic development. Thus, this simple kinetic alteration can indicate subfertyl cells, if considering molecular pathyways to embryo development.
My best regards!
Author Response
Response to Reviewer 1 Comments
Point 1: Lines 64-67: Spermatozoa analysis, but without mitochondrial test. If the authors putting all evaluations in this section, can be include this test.
Response 1: This comment has been taken into account (lines 66-67 ).
Point 2: Lines 73-79: The results are not significant? There aren't "p" description. If no significant, put, please, p >0.05.
Response 2: The "p" description was added In the chapter “results” (lines 73-79 ).
Point 3: Line 140: I didn't understand this phrase.
Response 3: It has been corrected (lines 140-142).
Point 4: Line 216: The authors used "fresh sperm". Fresh sperm, in our knowledge, it's a recent semen collected. After collection, dilution, and temperature reduction, this sample is cryopreserve, no?
Response 4: I agree with this suggestion. The "fresh sperm" has been replaced with "stored sperm one day"(lines 218-219).
Point 5: Line 220: repetitive expression. Our suggestion: "This may be due to individual differences among females deer."
About this point, my sight is the fertilization is a multifactorial condition that depend too oocyte quality, spermatozoa quality and, a third factor - quality of AI service (since hygiene procedure until manipulation stress).
Response 5: This suggestion was taken into account in the work (lines 222-223).
I agree that fertilization can be influenced by many different factors, and in addition to the quality of oocytes and semen, it is also necessary to take into account those related to the in vitro and in vivo procedures. With regard to AI, the correct recognition of heat and the correct conduct of the procedure at the right time has a significant impact on the effectiveness of insemination. In addition, the risk of microbial infections or the stress factor accompanying females during the AI procedure should be taken into account. Both factors can negatively affect the effectiveness of fertilization, the development of embryos and foetuses, and may also contribute to miscarriages. In the case of the hinds in this study, we observed that these animals, compared to other farm animals, were extremely sensitive to touch, which made the AI procedure much more difficult.
Taking into account the complexity of the factors that influence fertilization, we also included the indicated factors in the discussion (lines 254-257).
Point 6: Line 395: PMSG is a older term. Our suggestion is remove this information.
Response 6: This suggestion was taken into account in the work (lines 401-402).
Point 7: Line 425: The authors related: "Embryos (blastocysts) were collected between day 6 and day 9." This phrase connote that embryos were collected, probably, in the in vivo process. However, the embryos were cultivated in vitro and selected between daty 6 and 9 for transfer to deer females.
Response 7: Collection of embryos on the stage of blastocyst is typical for the purpose of checking its development rate. Usually it takes from 6 to 9 days (for sheep, bovine and deer embryos` culture). We didn`t provide AI, we checked the development rate of blastocysts.
Point 8: Line 447: one day or two day?
Response 8: Each hind treatment group was inseminated in accordance with the recommended insemination practice. The first treatment was performed 56 hours after the administration of eCG to the hinds (in the evening time) with sperm stored for 1 day, 7 and 10 days in groups 1, 2 and 3, respectively. The second treatment was performed 12 hours later in the morning with sperm from the same of stored samples, but time has already indicated that these sperm were stored for 2, 8 and 11 days, respectively. Therefore, in the methodology, such information is provided (lines 451-453). Additionally, this information is included in the diagram (Figure 2).
Point 9: And, in addition, in my perception about the kinetics parameters, the reduction of VSL and maintenance of VCL indicate probably spermatozoa circular movement, which may be a signal to reduction of sperm function, i.e., there are fertilization but decrease on embryonic development. Thus, this simple kinetic alteration can indicate subfertyl cells, if considering molecular pathyways to embryo development.
Response 9: I agree with the above observation that mobility disturbances related to kinetic parameters of movement significantly influence fertilization. Many studies on various animal species have shown significant relationships between VCL, VAP, VSL, ALH with fertility both in vitro and in vivo (Dorado et al. 2013, Holt et al., 1997, Love 2011, Silva et al., 2006,). In our experiment, the decrease in VSL and VAP while the maintaining of high VCL values showed a significant disturbance in sperm movement, the dominance of circulation, which could also affect fertilization efficiency and embryo development. Previous studies have shown that VSL may play a role in sperm transport through the reproductive system and in oocyte penetration (Gillan et al., 2008). On the other hand, the relationship between VSL and the pregnancy rate is not always confirmed (Dorado et al. 2013). It is difficult to clearly define how kinetic movement parameters influence embryo development, as it depends on various factors. Therefore, this issue remains to be clarified.
Referring to the above perception, a slight correction was included in the discussion (lines 166-168).
Thank you for taking the time to review the manuscript, and for all valuable remarks.

Reviewer 2 Report
The study intended to verify if red deer semen can be kept in specific medium culture, refrigerated and still maintain its fertility potential.
This study has its merit specially because the authors suggest that the semen can be kept in medium culture for until 10 days and still fertilize the egg.
Below there are some suggestion to help to improve the manuscript before publication:
1. A figure with study design the methods, would help the readers to picture the method contest.
2. In table 1, in the %of HD, the letters to demonstrate the difference among the groups go from a to c. There is no b. Could you explain that?
Still in table 1, what is the point in demonstrating Yo-Pro positive and negative if they are complementary from each other?
In results section, the authors state that oocytes recovered from the females were: 115 for day 1, 83 for day 5 and 44 for day 10. Could you explain such difference? Even if no difference was observed among fertilization rates, pregnancy rates, etc. this difference creates a bias in the study. It should be added in the discussion section a sentence about that.
Author Response
Response to Reviewer 2 Comments
Point 1: A figure with study design the methods, would help the readers to picture the method contest.
Response 1: A drawing showing the design of the experiments was prepared and attached to the paper (Figure 2).
Point 2: In table 1, in the %of HD, the letters to demonstrate the difference among the groups go from a to c. There is no b. Could you explain that?
Response 2: The percent (%) of HD were checked and the correction was carried in Table 1.
Point 3: Still in table 1, what is the point in demonstrating Yo-Pro positive and negative if they are complementary from each other?
Response 3: These parameters do not complement each other and are therefore presented. The study showed 4 types of populations according to the staining models presented. Only two populations of normal and apoptotic sperm are shown in the table. The populations of dying and dead spermatozoa are not presented.
Point 4: In results section, the authors state that oocytes recovered from the females were: 115 for day 1, 83 for day 5 and 44 for day 10. Could you explain such difference? Even if no difference was observed among fertilization rates, pregnancy rates, etc. this difference creates a bias in the study. It should be added in the discussion section a sentence about that.
Response 4: We tried to explain this aspect already in Discussion (Lines 220 – 223).
Thank you for taking the time to review the manuscript and for the positive feedback.

Reviewer 3 Report
In this study, Dziekońska et al. have provided a comprehensive analysis on the quality of red deer preserved in liquid state, both in vivo as well as in vitro. The study is quite extensive, and I appreciate that besides standard sperm characteristics, the fertilization potential was also assessed by IVF and insemination. In this sense, I regard this study as complex and insightful.
The manuscript reads well and cites appropriate references, , nevertheless, I would highly recommend to increase the visibility of Figure 1, since the graphs are very difficult to read.
Also, the authors noted a significant decrease of motility, VAP or VSL already on Day 5, while other parameters were impacted during later stages of storage. Any hypotheses on this observation?
On a minor note - please, remove the double parenthesis on Page 7, line 280.
Author Response
Response to Reviewer 3 Comments
Point 1: The manuscript reads well and cites appropriate references, , nevertheless, I would highly recommend to increase the visibility of Figure 1, since the graphs are very difficult to read.
Response 1: Figure 1 has been enlarged, and its readability will be clearer after the formatting process of the manuscript.
Point 2: Also, the authors noted a significant decrease of motility, VAP or VSL already on Day 5, while other parameters were impacted during later stages of storage. Any hypotheses on this observation?
Response 2: These studies suggest that the process of storing epididymal deer sperm at 5 ˚C causes the fastest changes in motion kinetics. A significant decrease in VSL while maintaining high VCL values indicates the presence of a significant subpopulation of sperm exhibiting a hyperactivated-like motility pattern (Suarez et al. 1992). Such a phenomenon was also observed in our earlier studies (Dziekońska et. 2020). The presence of a subpopulation of sperm showing hyperactivated-like motility pattern may also indicate a lack of full sperm maturation (Suarez et al. 1992), which may also affect their fertilization capacity.
Point 3: On a minor note - please, remove the double parenthesis on Page 7, line 280.
Response 3: Done as required.
Thank you for all comments and preparation of the review.
